# DICOMization of Proprietary Files Obtained from Confocal, Whole-Slide, and FIB-SEM Microscope Scanners

**DOI:** 10.3390/s22062322

**Published:** 2022-03-17

**Authors:** Yubraj Gupta, Carlos Costa, Eduardo Pinho, Luís Bastião Silva

**Affiliations:** 1Department of Computer Engineering, University of Aveiro, 3810-193 Aveiro, Portugal; carlos.costa@ua.pt; 2BMD Software, 3830-352 Aveiro, Portugal; eduardopinho@bmd-software.com (E.P.); bastiao@bmd-software.com (L.B.S.)

**Keywords:** pathogen niche, microscopy imaging, interoperability, DICOM, PACS

## Abstract

The evolution of biomedical imaging technology is allowing the digitization of hundreds of glass slides at once. There are multiple microscope scanners available in the market including low-cost solutions that can serve small centers. Moreover, new technology is being researched to acquire images and new modalities are appearing in the market such as electron microscopy. This reality offers new diagnostics tools to clinical practice but emphasizes also the lack of multivendor system’s interoperability. Without the adoption of standard data formats and communications methods, it will be impossible to build this industry through the installation of vendor-neutral archives and the establishment of telepathology services in the cloud. The DICOM protocol is a feasible solution to the aforementioned problem because it already provides an interface for visible light and whole slide microscope imaging modalities. While some scanners currently have DICOM interfaces, the vast majority of manufacturers continue to use proprietary solutions. This article proposes an automated DICOMization pipeline that can efficiently transform distinct proprietary microscope images from CLSM, FIB-SEM, and WSI scanners into standard DICOM with their biological information maintained within their metadata. The system feasibility and performance were evaluated with fifteen distinct proprietary modalities, including stacked WSI samples. The results demonstrated that the proposed methodology is accurate and can be used in production. The normalized objects were stored through the standard communications in the Dicoogle open-source archive.

## 1. Introduction

Following the discovery of single-celled organisms by Antonie van Leeuwenhoek [1] using his hand-crafted microscope, the use of microscopy in biological research has evolved greatly with the use of modern technology that has drastically enhanced microscope imaging capabilities. For example in the case of light microscopes, fluorescence microscopes allow for the identification and localization of fluorescent molecules in the given sample, whereas confocal microscopes which are an extended version of fluorescent microscopes allow for the acquisition of z-depth layer images of the same samples (if they have any) with higher resolution and later reconstruct those slices into 3D images [2,3].

Furthermore, since the advent of high-resolution digital microscopes over the decade, digital images have become one of the most significant groupings of linked data in automated analysis of various information related to biological structures and activities of living organisms which as a result shows studies on cell biology via cellular imaging have gained steadily increasing importance [4]. In addition to digital images, fluorescent probes and an electron and light beam were the other two astounding advances that have led to this increase. Likewise, the importance of cell imaging in pathogen niche research has grown significantly and this includes investigations on living cells such as cell phase identification, cell tracking, and tracking of subcellular components [5,6,7].

Despite the fact that microscope technology has made great advances in the field of life science, still, this sector faces substantial challenges in data processing and sharing due to the many digital file formats that are employed [8]. Furthermore, despite the fact that the functioning principles of numerous scanners are equivalent (like applying compression function, etc.), there are hundreds of microscope scanners on the market today that employ distinct digital file formats [4]. There are no universally accepted file formats for microscopy images. Many open-source and commercial software solutions for reading these diverse proprietary digital file formats have been created in order to mitigate the issue of distinct digital file formats [9,10,11]. These software programs differ in terms of their targeted application, usability, and source code accessibility [8]. Furthermore, the wide range of open-source and commercial solutions available today for determining which tool is ideal for a certain task may be tricky. Additionally, these solutions only provide access to image pixel data; sample clinical context (and acquisition process) information remain inaccessible [12]. As more competing vendor solutions emerge, the number of proprietary formats grows, posing a barrier to interoperability and maintainability. As a result, the open-source solution developers struggle to bridge the gaps between several proprietary formats, while the open-source image analysis software developer and commercial microscopy companies frequently engage in format conflicts [8,13]. Therefore, there is a strong demand in the microscopy sector for data standardization in order to improve clinical integration and support the computational development streams [4].

In the 1990s, different digital medical imaging modalities which were used in the hospital premises faced similar interoperability issues but after the acceptance of the Digital Imaging and Communication in Medicine (DICOM) standard protocol for data storage and communication that reality was transformed [14]. At the initial phase, DICOM gateways were required to integrate legacy systems and then convert their analogue media and proprietary digital formats into the standard DICOM and later integrate them with the institutional Picture Archiving Communication System (PACS) which facilitates the features of study sharing with other institutions for visualization or analysis [15]. Figure 1 shows the advantages of converting proprietary file formats into the standard DICOM. Aside from holding the actual image data, also known as pixel data, a DICOM object can also include a wide variety of metadata as well as properties that are part of the standard’s communication layers, thus enabling numerous services such as query and retrieval, data storage, acquisition scheduling, security profiles and print management [16]. Therefore, it is rightful to say that “DICOM standardization has completely addressed the issue of interoperability in the field of radiology”.

In this paper, we offer a conversion pipeline based on the standard DICOM environment that can efficiently convert several microscope imaging modalities from different scanners into the standard DICOM from proprietary imaging file formats that were gathered from confocal laser scanner microscope (CLSM), whole side imaging (WSI), and focused ion beam scanning electron microscopes (FIB-SEM) [17,18,19]. Later for validation reasons, the system was connected with the Dicoogle open-source PACS [20]. As a result, a vendor-neutral archive was created with the ability to receive studies from various equipment manufacturers and microscopy modalities, index and validate the accompanying information, streamline web-based viewing services, and provide access to third-party DICOM compliant applications.

## 2. Materials

The purpose of this study is to provide an automated pipeline for the DICOMization of multivendor CLSM, WSI, and FIB-SEM images with or without stacked imaging modalities obtained by cutting-edge scanners utilizing the methodological stages depicted in Figure 2.

### 2.1. Datasets

Data used in the preparation of this article were gathered from public bioimage data repositories as well as our partner institutes’ bioimage data repositories. In the case of CLSM microscope, .czi, .nd2, and .lif imaging modalities extension files were provided from our partner institute and .lsm imaging modalities were downloaded from The Cell: An Image Library [21]. Likewise, in the case of WSI imaging modalities, stacked .ndpi image extension files were provided from our partner institute and remaining .scn, .svs, .tif, .tiff, .bif, .vsi, and ome.tiff imaging modalities were downloaded from Cytomine [22], TCGA [23], and Openslide-testdata [24] public bioimaging repositories. Moreover, In the case of FIB-SEM imaging modalities, a series of .tif images were provided from our partner institutions and the remaining .mrc imaging files were downloaded from the Electron Microscopy Public Image Archive (EMPIAR) [25].

### 2.2. Microscope Systems

The microscope system enhances the viewer’s resolution (the power to discern two nearby objects as distinct objects), contrast (the power to differentiate different sections of the sample based on color or intensity), and magnification (the power to see the specimen at a higher magnification) [26]. The human eye can distinguish objects with a resolution of 0.1 mm, whereas a light microscope with a magnification of 1000× can resolve objects with a resolution of 0.2 mm (200 nm). Objects as tiny as 0.1 nm can be detected with an electron microscope. In general, microscopes employ refraction or reflection, dispersion of electron beams/electromagnetic radiation interacting with materials to provide a signal to form an image or a collection of dispersed radiation. In the context of cellular organisms and their components, which are often quite minute, exploration requires the use of a microscope with a suitable resolution, contrast, and magnification. In this paper, we present a DICOMization architecture for three demanding modalities in pathogen niche research: Confocal laser scanner microscope (CLSM), brightfield microscope (or whole slide image), and focused ion beam scanning electron microscopy (FIB-SEM).

#### 2.2.1. Confocal Laser Scanning Microscopy

Confocal microscopy is one of the most widely utilized imaging modalities in the biological sciences today, and despite its high cost, its use and applications are expanding day by day in both the biology and medicine fields [3,27]. For the thin tissues, widefield epifluorescence microscopy is a preferable choice and is a straightforward and quick approach to capture all illumination emitted light and image the specimen, but in the case of thicker cells, out-of-focus light may obscure fine structures in the objective focal plane. This problem is avoided in CLSM by including a physical pinhole coupled to the focal plane to exclude out-of-focus light reaching the eyepieces or detector. This generates an optical sectioning effect, allowing users to capture just light from the objective’s focal plane. It also gives “pseudocoloured” images that correlate to the channels that have been assigned. Lately, all recent CLSM designs have been based on a standard upright or inverted research-level optical microscope. The CLSM device can obtain successively high contrast images along the focal length or z-axis of the sample to depths in the range of 10–50 μm. This stacked image (representing different focal planes) can be visualized in a three-dimensional view with the help of computer software, letting the samples be viewed in several planes.

For this article, we have collected four different proprietary file formats of CSLM imaging modalities: .czi and .lsm were obtained from LSM780 and LSM710 scanner (belongs to Carl Zeiss lab), .nd2 obtained from Nikon A1 confocal microscope scanner, and .lif imaging modalities were obtained from DMI8-CS scanner belongs to Leica Lab. Figure 3a shows the six-channel CLSM image generated by Zeiss scanner and has an extension as .czi.

#### 2.2.2. Whole Slide Imaging

Whole-slide imaging is the process of scanning a whole microscope glass slide and generating a single high-resolution pyramid structure digital file. This is generally accomplished by capturing many small compressed high-resolution tiles (through applying tile-by-tile or line-scanning approach), which are then stitched together to form a complete digital image of a histology section [28]. This obtained high-resolution digital data can be effectively stored, retrieved, analyzed, and disseminated via the web to different research institutes. The captured tile images are usually compressed into JPEG, JPEG 2000, or LZW compression to reduce file size and afterwards stitched and saved in a proprietary file format (generally implemented on the notion of TIFF/BigTIFF-baseline characteristics [29]). Commercial WSI scanners use diverse proprietary file formats that require specialized software to read and analyze. This signifies that there is no widely accepted standard file format for this image. A single whole slide image may be scanned at multiple magnification levels. For regular hematoxylin-eosin (H&E) and immunohistochemistry (IHC) slides, scanning at 20× magnification is generally sufficient but for other applications, such as digitization on in situ hybridization slides, images should be acquired at 40× magnification level to resolve information that may be separated by distances less than about 0.5 μm. WSI z-plans are imaginary focal layers that depict the image at a specified physical height (in microns) over the reference layer (which is the top surface of a glass slide).

For this article, we collected nine different proprietary file formats of WSI imaging modalities: stacked .ndpi images were obtained from C9600-12 scanner (belongs to Hamamatsu lab), .svs obtained from Aperio scanner, .scn obtained from Leica scanner, .bif files were obtained from Ventana scanner, .vsi were obtained from Olympus scanner and remaining extension files .tif, .tiff, and OME.tiff were downloaded from Cytomine and Omero public repository sites. Figure 3c shows the sample of WSI image generated by Hamamatsu scanner and has an extension as .ndpi.

#### 2.2.3. Focused-Ion Beam Scanning Electron Microscope

The FIB-SEM working principle is similar to the Scanning Electron Microscope (SEM), but instead of utilizing a focused beam of the electron over the specimen surface it uses focused ions, typically gallium ($Ga^{+}$)as the ion source, to image the specimens. Modern FIBs and SEMs are merged into a single device that allows in situ ion milling and imaging using either beam [30,31,32]. Both FIB and SEM can be used independently, but combining them into a single layer opens a wider range of options that are otherwise not possible. This hybrid equipment is known as a dual-beam FIB or a FIB-SEM. This merged device is very beneficial for cross-section specimen preparation since the electron beam can observe the cross-sectional face while the ion beam etching (or ion beam mills) can accomplish a well-prepared specimen surface image of high-resolution for image analysis. Ion beam mills aid in the removal of leftover artefacts that may be present during the mechanical cutting and pollination of samples. Ion beam polished cross-section specimens produced by ion beam mills can be used for electron microscope imaging. The primary distinction between TEM/SEM and FIB is the utilization of ions as the beam and also for the interactions that occur at the specimen surface.

For this article, we collected two different proprietary file formats of FIB-SEM imaging modalities: .tif extension file was provided by our partner institutions and the remaining .mrc were downloaded from the Electron Microscopy Public Image Archive (EMPIAR). Figure 3b shows the sample of FIB-SEM image generated by electron scanner and has an extension as .tif.

### 2.3. Digital Imaging and Communication in Medicine (DICOM)

DICOM is a widely recognized standard protocol for the storage and communication of imaging data, digital file format, and image file hierarchy, as well as the storage of image-related information. For medical images and image-related data, it specifies a nonproprietary digital image format, data transfer protocol, and file structure [16,33]. Hospitals and medical institutes have adopted it, and it is making inroads into smaller devices to physicians’ offices. It also allows for the integration of PACS with acquisition equipment, workstations, printers, servers, and network gear from a variety of manufacturers. The DICOM standard demonstrates how to organize and exchange medical imaging modalities and associated data both within and outside the institution (e.g., telemedicine, teleradiology). The standard has been kept current by the publication of supplements that provide interoperability with newer acquisition equipment and technologies.

DICOM-formatted medical images are binary files made up primarily of two logical sections: A metadata header and image data (pixel data). A DICOM image file is made up of a file header and image datasets that are compressed into a single file. DICOM files are made up of a large number of DICOM data elements, and each data element contains meta-information about the image, such as patient and clinical staff information, acquisition equipment parameters, radiation dosage, and structured reports. A single frame DICOM object has only one attribute containing pixel data; likewise, for many modalities, this resembles a single image. However, sometimes the attributes may contain multiple “frames”, allowing storage of other multi-frame data.

DICOM data elements are arranged into modules known as Information Object Definition (IOD). An IOD is an object-oriented conceptual data model that is used to identify information about real-world objects such as modalities or patient-specific features. DICOM standard committees decide and determine which IOD is assigned to specific modalities. Likewise, the transfer syntax element is used to encode the image dataset (this describes the object structures and data encoding, for file storage and network transfer purposes), which must be the one provided by the Transfer Syntax UID class of the DICOM standard File Meta Information. The transfer syntax for DICOM files is stored in the file metadata header, whereas it is negotiated between the service class provider and the service client user for networking. The DICOM standard key benefit is that it allows for interoperability (industrialization of microscopy imaging) with different systems and DICOM-compatible equipment. DICOM also has a working group for digital pathology (Working Group 26) that has already released two additional DICOM supplements, 122 and 145. The 145 supplement is concerned with WSI archiving, whereas supplement 122 is concerned with the method of pathology specimen identification.

## 3. Overview of the Proposed Framework

The proposed framework, seen in Figure 2, consists of three conversion pipelines that convert various proprietary file formats received from CLSM, WSI, and FIB-SEM microscope scanners into DICOM format by following the standard DICOM protocol.

### 3.1. CLSM and FIB-SEM Conversion Pipeline

Figure 4 depicts the DICOMization pathway for the CLSM and FIB-SEM imaging modalities. The conversion pipeline consists of two main processing steps. The first is to choose an image file reader based on the provided proprietary files given. In our experiment, we used the open-source OME-Bioformats library to read several proprietary files generated from microscope scanners. According to the OME-Bioformats website https://docs.openmicroscopy.org/bio-formats/6.8.1/supported-formats.html (accessed on 20 February 2022), their library presently supports 159 different formats. While OME-Bioformats code is written in java, with the help of Java-bridge, we can use their library in a Python environment. Furthermore, in the second stage, the Pydicom 2.0.0 library is utilized to transform provided proprietary files to the standard DICOM. This library reads and writes DICOM files in a Python environment very quickly. It will generate new dataset main object files, which will contain file meta information for dictionary files such as *Media Storage SOP Class UID*, *Implementation Class UID*, *Transfer Syntax UID*, and so on.

In terms of image structure, FIB-SEM scanners create serial 2D images of the provide samples, but CLSM scanners may produce single, double, triple, or multi-wavelength illumination mode images from the provided sequence of individual optical sections, which is a basic image unit. Later, created multi-wavelength mode images are blended with one another to observe all imaging modes in one; each mode is also referred to as a channel. In addition, each channel is acquired at a certain time interval with a sequence of z-series data of individual optical sections. So, in general, the dimensions of CLSM images are on the order of four (XYCZT).

In terms of image conversion, proprietary files of FIB-SEM and CLSM imaging modalities [17,18] were first sent to the OME-bioformats library, which then chose an appropriate image reader depending on the provided sample. If the OME-bioformats library recognizes the files, it will read the image pixel data and the metadata of the given sample. Later, this gathered data will then be sent to the Pydicom library. Before converting the acquired 2D slices from the first phase into DICOM, we must first construct new dataset main object files that encompass the dictionary files such as *File Meta Information Group Length*, *Media Storage SOP Class UID*, *File Meta Information Version*, *Implementation Class UID*, *Media Storage SOP Instance UID*, and *Transfer Syntax UID*, as well as file meta information. Moreover, we selected the Class UID module as 1.2.840.10008.5.1.4.1.1.77.1.2 *(VL Microscopic Image Storage)* and 1.2.840.10008.1.2.1 *(Explicit VR Little-endian)* as a Transfer Syntax UID for both imaging modalities (since the provided CLSM and FIB-SEM images were uncompressed, furthermore, the pipeline could be easily adjusted to use another encoding, as a trade-off between image quality, data size and conversion time). The “MONOCHROME2” option was chosen for Photometric interpretation attributes since the given sample was grayscale with a sample per pixel of one. In the following step, we created some private tags to fill the newly created DICOM image metadata with microscope image information such as the channel name, image dimension order, illumination types, pixel size, channel ID, magnification range, and so on, because the DICOM normalization committee’s defined DICOM file header does not provide public tags to insert this type of information inside the metadata. Similarly, the SHA-1 hash function library was used to establish a unique patient ID for each and every sample. Our pipeline can transform both 8-bit and 16-bit imaging modalities into standard DICOM according to standard tags (*Bits Allocated*, *Bits Stored*, *High Bit*).

### 3.2. WSI Conversion Pipeline

Figure 5a illustrates the DICOMization pathway for the WSI imaging modalities. The proposed WSI conversion pipeline is divided into three stages: Image file reader and unrecognized file conversion to the OME.tiff file format; the internal processing stage includes multiple steps such as image series count, channel count, z-series layer count, splitting the image into a number of tiles, pyramid layer extrapolation and others, and DICOMization stage.

The conversion approach starts by sending the original WSI samples to the OME-Bioformats file reader, which decode image pixel data and their metadata if the file format is recognized [19]. If the provided WSI file format is not recognized by this library, then the proposed pipeline will use a different process, activating the OME-tiff function, which will convert an unrecognized file to a pyramidal OME-tiff file by reading image tiles with the TIFFFile Python library and then stitching these tiles back together to reconstruct the original WSI image into pyramidal OME.tiff file format with their metadata.

Following the successful decoding of the sample by the OME-Bioformats file reader, the pipeline counts the number of resolution levels (or image series count) contained in the original image, followed by a count of z-series layers to decide if the passed image contains a z-axis plane or not. If the provided sample lacks a z-series plane, then our pipeline will bypass the z-stack loop and send the acquired image (x and y) coordinates from the first step (image series count) to the function-1 (process-1) and function-2 (process-2) without z-stack loop for further processing. After finishing the first layer’s processing, the process will return to the resolution layer count loop to load the next layer. This operation will be repeated until all resolution layers have been processed from functions (1 and 2). Furthermore, if the provided sample had z-series data then our pipeline will bypass without z-stack loop and send the collected image (x and y) coordinates of the first z-series from the first step (image series count) to the function-1 (process-1) and function-2 (process-2) of with z-stack loop for further processing. This loop will repeat over each z-axis resolution layer while simultaneously passing each z-depth resolution layer via function-1 and function-2 for additional processing. After it has completed running for each z-series of the first image series count, it will return to the image series count loop to read another resolution layer; this process will be repeated until all stacked resolution layers have been passed from function (1 and 2) for conversion into standard DICOM.

Overall, to address the zoom gap problem between resolution layers that often happens in WSI images as shown in Figure 6a,b, we designed two functions: function-1 and function-2, in the internal processing stage. Function-1 operates by dividing the input image into a number of small tiles, the size of which is determined by the user. In our pipeline, we have set the tile size to be 512 × 512. After cropping a passed input image into a number of tiles (the number of tiles depends on the size of the resolution layer, it varies within resolution layers, as shown in Figure 7); these cropped tiles will be appended one by one to form a list of tiles, which will then be passed to the DICOM encapsulate function to create an encoded multi-frame DICOM image. Like function-1, function-2 will crop the provided input image in 1024 × 1024 dimensions, which is twice the cropped dimension of function-1, and then down-sample (or resize) these cropped tiles by dividing their width and height by two, as illustrated in Figure 5b. Later, it will add each tile into a single form for multi-frame encoding.

While converting stack or non-stack WSI images into standard DICOM, we utilized the SHA-1 function from the hashfile library to generate a unique identifier for each sample. As a result, the newly generated DICOM files will contain unique UIDs for series, study, and SOP instance (tags *SeriesInstanceUID*, *StudyInstanceUID*, and *SOPInstanceUID*). Furthermore, we passed *JPEG baseline* (1.2.840.10008.1.2.4.50) as the transfer syntax for serialized multi-frame encoding of multi-frame DICOM images. Similarly, for the *SOP Class UID*, we provide virtual light (VL) whole slide microscopy image storage (1.2.840.10008.5.1.4.1.1.77.1.6), which was published by DICOM committee members. The YBR_FULL_422 option was chosen for photometric interpretation attributes since the given sample was RGB with a sample per pixel of three. Later, we generated some private tag dictionaries to fill the information collected from the provided samples, such as *Device Maker*, *Capture Mode*, *Device Model*, into the newly formed DICOM image metadata.

## 4. Result and Discussion

Here, we intend to build an automated process that can convert a variety of microscope proprietary data into standard DICOM format, which would address not just interoperability but also sharing, maintainability (across institutions), and visualization. The Python environment was chosen as a programming platform for this project due to its versatility and available resources. The above-mentioned pipeline was performed on an Intel Core i7 10th generation CPU with 16 GB RAM and the Ubuntu 18.04 LTS operating system.

In this section, we will present the findings obtained after evaluating the performance and reliability of our proposed pipeline for converting distinct proprietary file format inputs into standard DICOM as input samples were gathered from distinct scanners belonging to CLSM, FIB-SEM, and WSI microscope systems.

As shown in Table 1, we collected four distinct proprietary file formats belonging to the CLSM microscope. The Zeiss scanner has two files extension, .czi and .lsm, but the Nikon and Leica scanners have one each, .nd2 and .lif. The obtained .czi and .lsm samples were 16 and 8 bits per pixel, respectively, whereas .lif and .nd2 were 8 and 12 bits per pixel, showing that the CLSM pipeline was tested with a varying range of bits per pixel samples.

Furthermore, the number of channels and image size of the gathered input samples varies, and each of them has a series of z-stack planes. In the instance of the .lif sample, which is a four-channel sample with 64 z-series planes on each channel, in total there are 256 image slices stacked on top of one another. To convert this into DICOM, our proposed pipeline took 9.56 s as the size of the Leica image was 900 MB. Similarly, with a .czi image with a file size of 431 MB and 138 slices in total, our proposed pipeline took 2.12 s to convert to the standard DICOM. Figure 8 shows the DICOMized results of all four selected proprietary files from CLSM microscopes.

Additionally, Table 2 shows the generated unique identifier for the .czi extension file. These unique identifiers were generated for each input sample while converting them to DICOM using the SHA-1 function from the hashfile library (which are globally unique values). As shown in Table 2, the newly produced DICOM file of .czi includes the following number in a patient ID tag: 186355337916212766286352571899677090176885900481. It also has distinct *StudyInstanceUID*, *SeriesInstanceUID*, and *SOPInstanceUID* values. Overall, in the instance of CLSM proprietary files, our proposed pipeline effectively converted passed distinct CLSM proprietary files into DICOM by following the standard DICOM protocol. Figure 9a shows the biological information of the first channel .nd2 image, which is stored inside the DICOM metadata.

In the instance of the FIB-SEM microscope, we obtained two distinct proprietary input samples, one with the .tif extension and the other with the .mrc extension, as shown in Table 3. Both files have an 8 bit per pixel resolution. The .tif input sample size was 2.7 GB and contained 447 series of 2D images, whilst the .mrc input sample size was 948 MB and contained 361 series of 2D images. To convert these two files into DICOM, our proposed pipeline took 14.76 and 5.74 s, respectively. Figure 10 depicts the DICOMized result and Figure 9b shows the biological information of .tif image, which is stored inside the DICOM metadata. As OME-bioformats was unable to read .mrc files, we first pass .mrc files from the mrcfiles library in Python to read the image array, and then we send those arrays to the OME-bioformats library internally to convert them to DICOM files.

To evaluate the conversion performance and reliability of the proposed pipeline for WSI samples, we utilized nine distinct proprietary files, two of which were stacked while the remaining sample files were non-stack (non-stack: .svs, .scn, .tif, .tiff, .ome.tiff, .bif, .vis, and stack: .ndpi). Furthermore, the OME-Bioformats library successfully read all nine provided proprietary files, including their image pixel data and metadata. It was chosen as a file reader for this pipeline over other open-source libraries such as openslide [10] because it can read both stack and non-stack image files.

Despite the fact that OME-Bioformats read all passed proprietary files efficiently, it could not always read the pyramidal resolution layer of .tiff and .tif files other than the base resolution layer. To overcome this issue, we build a function that uses the Tifffile Python module to convert these unrecognized resolution layer image files into pyramidal OME.tiff files. To test the function’s effectiveness of the above-mentioned OME.tiff conversion technique, we send two WSI files (.tif and .tiff, both downloaded from the cytomine website) to the above-mentioned technique whose files resolution layer is unrecognized by the OME-Bioformats image reader except for the base one. These are terabyte and gigabyte image files that have been compressed to 158 GB and 847 MB, respectively, using a JPEG code stream of quality 70 and 92 percent, as shown in Table 4.

Furthermore, it took 10 h for our OME.tiff conversion algorithm to convert 158 GB of sample data into pyramidal OME.tiff file and 2.72 min to convert 847 MB of image data into a pyramidal OME.tiff file. For this function, we encoded each image using lossy JPEG compression at 90% quality [34,35] (level 90 JPEG compression produces typically compression ratios of about 1:10 in pathology WSIs), so the first image size increased to 247.2 GB from 158 GB because it was originally compressed at 70%, and the second image size decreased to 713.5 MB from 847 MB because it was originally compressed at 92 percent quality. After completing the OME.tiff conversion, the transformed images were forwarded to the internal processing section for the conversion into standard DICOM, as shown in Figure 5a.

Additionally, as seen in Figure 6c with pink color, certain WSI samples had a resolution gap between each layer. For example, in the .ndpi samples obtained from Hamamatsu scanners, the base layer was zoomed at 40× and its size was 166,656 × 60,928 × 11, but the second layer was scanned at 10× and its size was 41,664 × 15,232 × 11, resulting in a zoom gap of 30×. These types of images can be DICOMized and visualized, but without the addition of more resolution layers, the viewer will need to request a significantly larger number of tiles for the next magnification level in order to show the contents of the viewport with high fidelity, but at a higher network performance cost. To address this issue, we proposed two functions as their workflow is already described in the *WSI Conversion Pipeline* section. We found the missing resolution gaps using the algorithms described above, as indicated by the green color in Figure 6c.

Table 5 shows the proposed method’s performance and conversion time for converting both stack and non-stack proprietary WSI sample files into standard DICOM. While encapsulating, each pixel data was encoded to the multi-frame using lossy JPEG compression at 90% [34,35]. As seen in Table 5, converting 11 stacks of WSI images into DICOM took more than 10 h. In addition, the resolution layer of the stacked images was increased from five to 10. Figure 11 shows the 11 stack .ndpi WSI pyramid image that has been successfully converted to standard DICOM and Figure 9c shows the biological information of the base layer of .ndpi (patient name: DCM_1D_1c-6c) WSI sample, which is stored inside the DICOM metadata. In the instance of the .ndpi WSI sample, which was an 11 stack image, after passing it through our proposed pipeline, as shown in Figure 5a, the internal process function and DICOMization algorithm will automatically produce 11 folders (folders depend on the number of stacks of the sample), each of which contains ten DICOMized images, as shown in Figure 11. Likewise, Figure 12 shows the DICOMized image of the first stack from the 11 stack .ndpi image.

In this experiment, to check whether the generated DICOM files were readable or not, we passed each DICOM file from the Pydicom library to read it in a Python environment, and we passed each generated DICOM file to the open-source PACS archive Dicoogle, to be opened by the PACScenter viewer. Figure 13 shows that all nine WSI, four CLSM, and two FIB-SEM DICOMized files were successfully indexed by the Dicoogle archive. The DICOMized result is available online [36].

## 5. Conclusions

In the present context, there are not any decentralized open-source applications that can provide interoperability solutions in the domain of microscope imaging since there is no universal standard for this. Therefore, in this scenario, we believe that adopting the universal DICOM standard is the best way to address interoperability and proprietary concerns.

As a result, the goal of this article was to develop an automated DICOMization pipeline that can take distinct proprietary microscope imaging files and efficiently convert them into standard DICOM with their biological information stored inside their metadata, allowing for interoperability between proprietary files as well as the flexibility to share and visualize in any local DICOM viewer. To test the performance and reliability of our proposed pipeline, we pass 15 distinct proprietary files, four of which belong to the CLSM scanner, two to the FIB-SEM scanner, and the other nine to the WSI scanner. Table 1, Table 3 and Table 5 show the outcomes. As illustrated in Figure 13, our proposed pipelines successfully converted all 15 distinct proprietary files into standard DICOM.

The number of successfully verified proprietary files in our study is modest due to a lack of publicly available data, so we could not evaluate our pipeline performance to the maximum height, but we believe it can convert all extension files supported by OME-Bioformats. Furthermore, in order to simplify the complexity and breadth of the results, the current study was restricted to the use of a baseline JPEG encoding for the compression of a number of tiles into a multi-frame DICOM in a WSI pipeline. Nonetheless, the application and analysis of different image encoding formats is an area of future study that should be pursued. In addition, we would like to test our pipeline with other microscope imaging modalities that are not presently included here (namely, .oib, .mirax, and .svslide). These methods are part of our vision towards the creation of an open-source N-dimensional viewer for modern microscopy DICOM images.

## Figures and Tables

**Figure 1 sensors-22-02322-f001:**
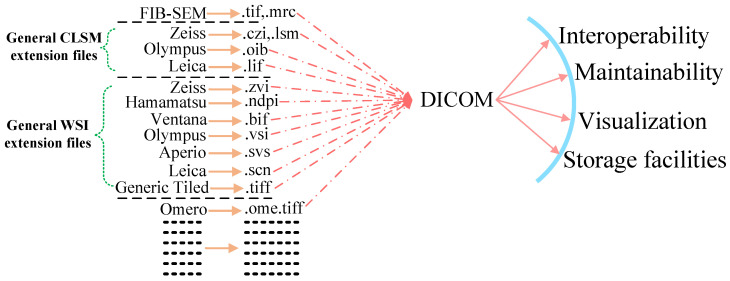
Depiction of the conversion of distinct proprietary files into DICOM, with advantages, enumerated.

**Figure 2 sensors-22-02322-f002:**
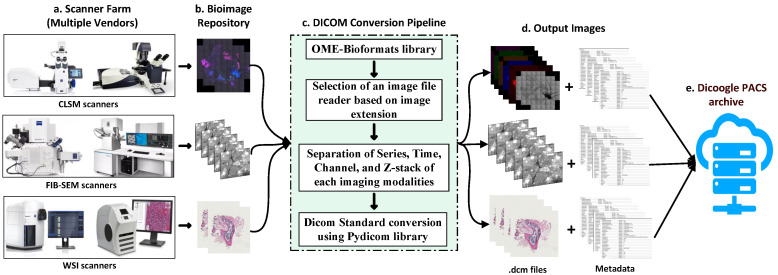
Overview of the proposed pipeline for the conversion of CLSM, WSI, and FIB-SEM microscope imaging modalities into the standard DICOM.

**Figure 3 sensors-22-02322-f003:**
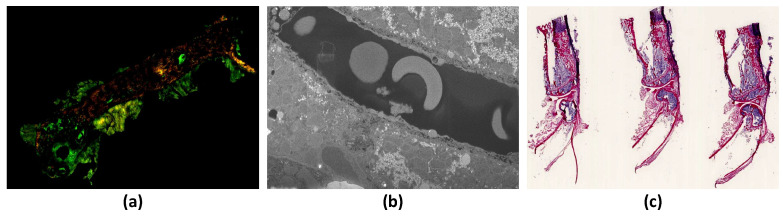
Samples of different microscope imaging modalities (**a**). Six-channel mouse bone tissue (left leg femur, infected treated unsuccessfully) (*S. aureus*: Visualized in blue; RUNX2: Visualized in pink; nuclei: Visualized in green; actin cytoskeleton: Visualized in red; collagen (SHG 2 photon): Visualized in orange): Generated by Zeiss scanner, with the .czi extension (**b**) FIB-SEM sample image showing liver microvili: Generated by electron scanner, with the .tif extension (**c**). WSI sample image of mouse bone tissue (left leg femur, infected treated unsuccessfully): Generated by Hamamatsu scanner, with the .ndpi extension.

**Figure 4 sensors-22-02322-f004:**
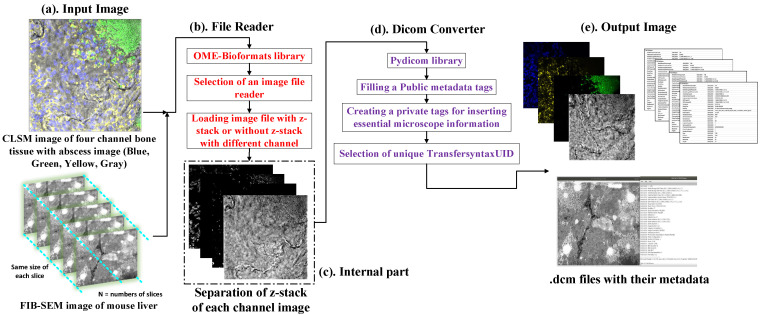
Pipeline for the conversion of CLSM and FIB-SEM microscope imaging modalities into standard DICOM.

**Figure 5 sensors-22-02322-f005:**
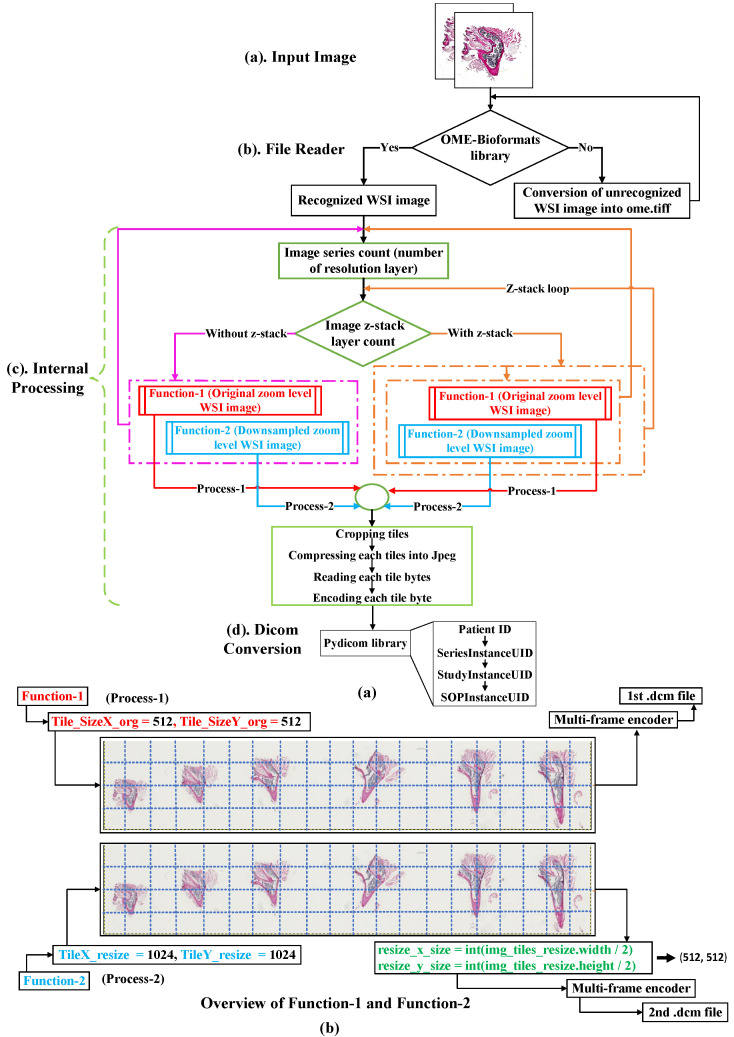
Pipeline for the conversion of WSI imaging modalities into standard DICOM (**a**) Workflow of the pipeline, (**b**) Overview of function-1 and function-2.

**Figure 6 sensors-22-02322-f006:**
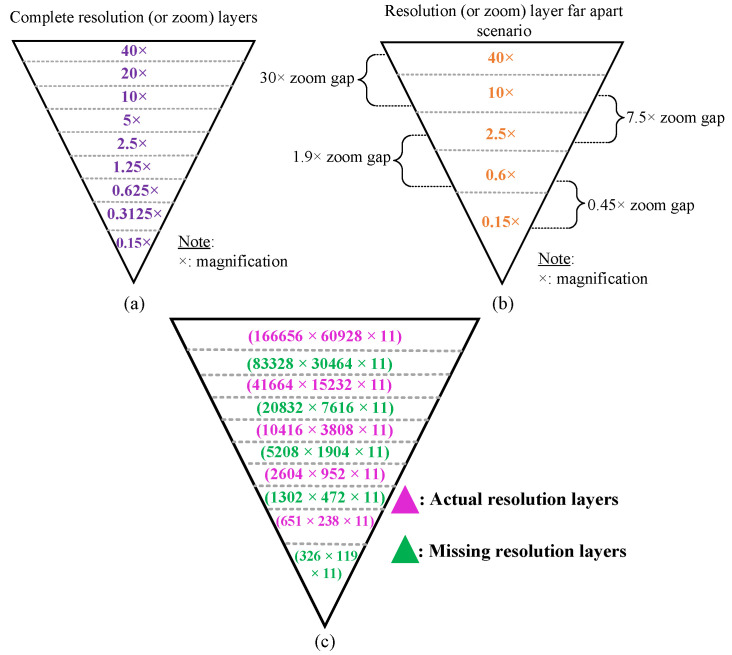
WSI image series of resolution layers; (**a**) complete resolution layers; (**b**) some missing resolution layers; (**c**) actual and found missing resolution layers after applying function-1 and function-2 for the .ndpi sample.

**Figure 7 sensors-22-02322-f007:**
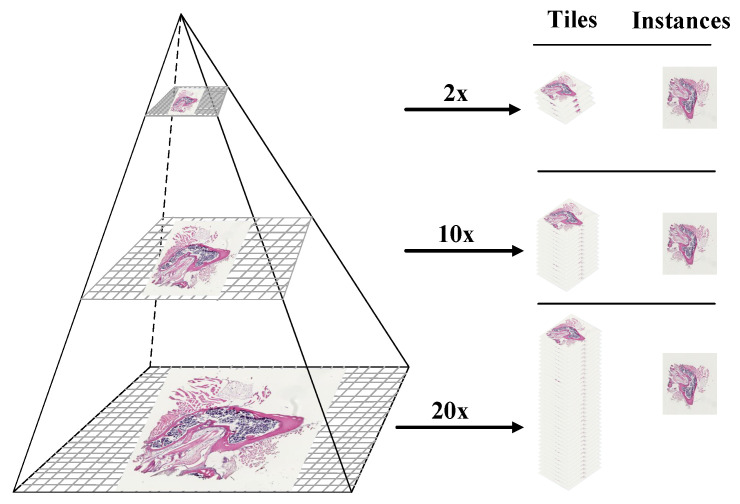
Mapping a DICOM-compliant multi-resolution WSI pyramidal image. Each layer of the pyramid is a downsample of the WSI picture and is made up of a series of tiles. The tiles are encrypted as separate frames of the multi-frame DICOM instances (files).

**Figure 8 sensors-22-02322-f008:**
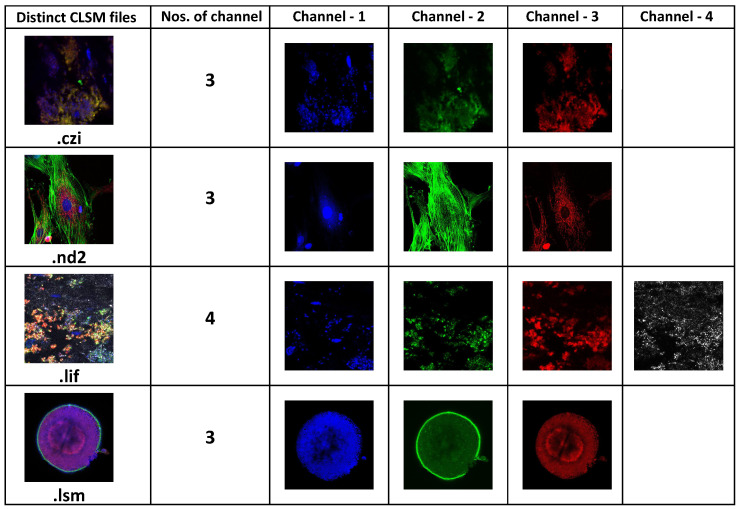
Four distinct (.czi, .nd2, .lif, and .lsm) CLSM microscope images successfully converted into standard DICOM.

**Figure 9 sensors-22-02322-f009:**
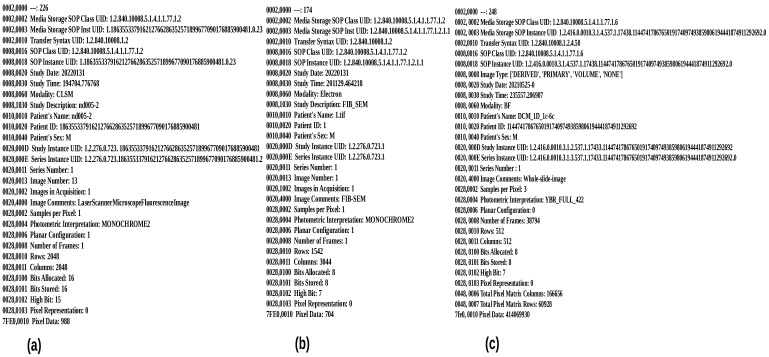
Bioimaging information of each microscope imaging modality is stored inside the metadata of each image: (**a**) Biological information of channel-1 of CLSM image (Nikon scanner), (**b**) FIB-SEM .tif sample biological information, (**c**) Metadata information about WSI sample (DCM_1D_1c-6c), base resolution layer.

**Figure 10 sensors-22-02322-f010:**
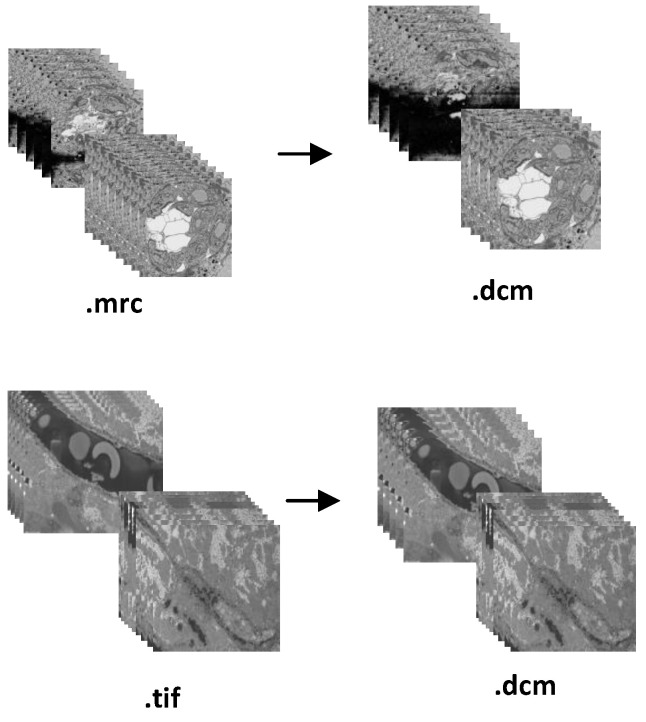
Two distinct (.mrc and .tif) FIB-SEM microscope images were successfully converted into standard DICOM.

**Figure 11 sensors-22-02322-f011:**
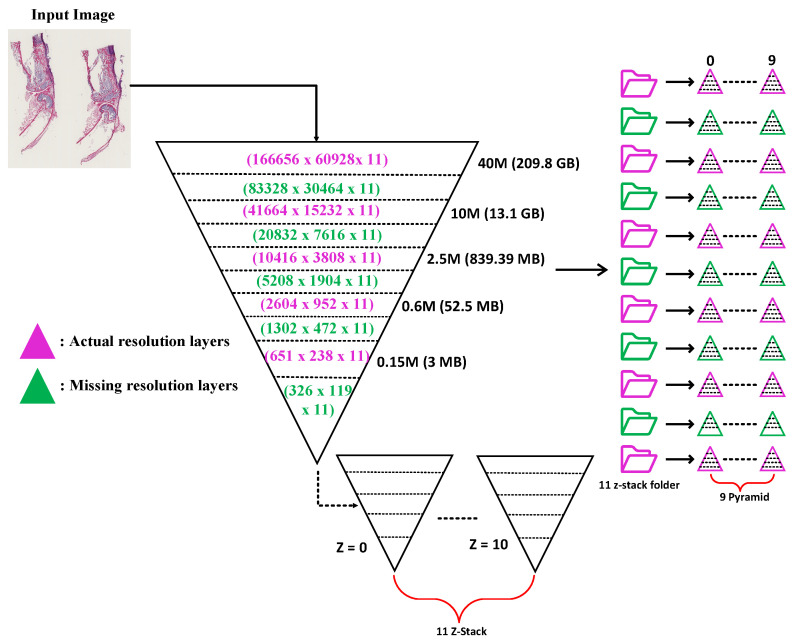
Eleven stack WSI images (belongs to .ndpi extension) successfully converted into standard DICOM.

**Figure 12 sensors-22-02322-f012:**
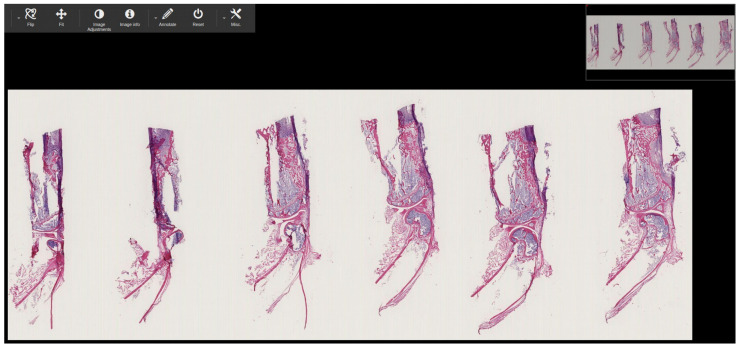
Screenshot of the generated DICOM file (of 11 stacks .ndpi WSI image) into Dicoogle WSI web viewer.

**Figure 13 sensors-22-02322-f013:**
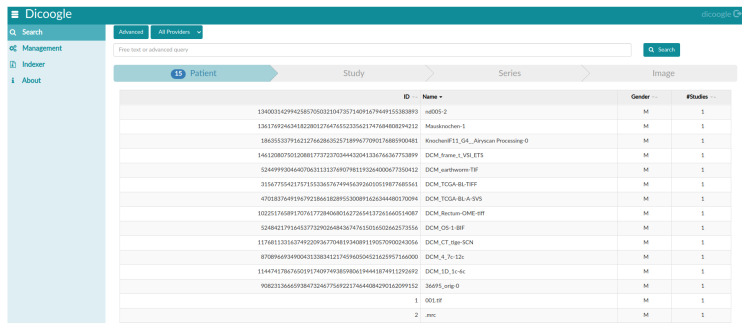
Dicoogle PACS shows that the files in the eight distinct proprietary file formats were successfully indexed by the archive.

**Table 1 sensors-22-02322-t001:** CLSM to DICOM conversion framework performance measured by passing four distinct proprietary files.

Distinct Scanners	File Extension	Bit Depth	Image Size	Nos. of z-Stack Image	Nos. of Channels	Total Image	Conversion Time (s)
Zeiss	.czi	16	431 MB	46	3	138	2.12
Nikon	.nd2	12	604 MB	24	3	72	7.39
Leica	.lif	8	900 MB	64	4	256	9.56
Zeiss	.lsm	8	708 MB	56	3	168	6.69

**Table 2 sensors-22-02322-t002:** Generated unique identifiers for the file originally with the .czi extension.

Tags	Unique Identifier
Patient ID	186355337916212766286352571899677090176885900481
Study Instance UID	1.186355337916212766286352571899677090176885900481
Series Instance UID	1.186355337916212766286352571899677090176885900481.0
SOP Instance UID	1.186355337916212766286352571899677090176885900481.0.23
SOP Class UID	1.2.840.10008.5.1.4.1.1.77.1.2
Transfer Syntax UID	1.2.840.10008.1.2

**Table 3 sensors-22-02322-t003:** FIB-SEM conversion pipeline performance of two distinct FIB-SEM image files into standard DICOM.

File Extension	Pixel Type	Image Size	Total Image	Conversion Time (s)
.mrc	8	948 MB	361	5.74
.tif	8	2.7 GB	447	14.76

**Table 4 sensors-22-02322-t004:** Conversion of unrecognized WSI image into OME.tiff.

File Extension	Image Size	Compression Quality	Converted to OME.tiff	Compression Quality	Image Size	Conversion Time
.tiff	158 GB	Jpeg: 70	1.ome.tiff	Jpeg: 90	247.2 GB	10 h
.tif	847 MB	Jpeg: 92	2.ome.tiff	Jpeg: 90	713.5 MB	2.72 min

**Table 5 sensors-22-02322-t005:** Proposed WSI framework performance while converting eight distinct proprietary WSI files into standard DICOM files.

File Extension	Actual Number of Resolution Layer	Image Dimension (x, y, z)	Stack (Y/N)	Image Size	Total Converted DICOM File Size	Total Number of Resolution Layer after Conversion	Total Conversion Time (min)
.ndpi	5	(166,656 × 60,928 × 11)	Y	7.3 GB	6.4 GB	10	581.65
.ndpi	5	(142,848 × 68,608 × 11)	Y	9.4 GB	7.9 GB	10	863.29
.scn	6	(41,984 × 41,088)	N	149.8 MB	183.2 MB	12	2.41
.svs	5	(18,9448 × 41,237)	N	871.6 MB	777 MB	10	19.95
.tif	9	(42,460 × 29,140)	N	141.6 MB	95.9 MB	10	2.18
.tiff	9	(46,000 × 32,914)	N	218 MB	153.9 MB	10	2.89
ome.tiff	10	(69,632 × 72,704)	N	713.5 MB	487.1 MB	11	9.64
.bif	10	(105,817 × 93,978)	N	3.9 GB	1.7 GB	11	109.32
.vsi	10	(66,982 × 76,963)	N	512 MB	982.3 MB	10	3.15

## Data Availability

The supporting result can be found here: https://figshare.com/projects/Dicomization_of_Proprietary_Files_obtained_from_Confocal_Whole-Slide_and_FIB-SEM_Microscope_Scanners/132077 (accessed on 20 February 2022).

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
