# Peer review of "DICOMization of Proprietary Files Obtained from Confocal, Whole-Slide, and FIB-SEM Microscope Scanners"

_sensors, 2022, doi:10.3390/s22062322_

Round 1

Reviewer 1 Report

The main objective of the work is to provide an automated conversion pipeline based on the DICOM standard environment, that can efficiently convert several microscope imaging modalities from different scanners (confocal laser scanner microscope, whole side imaging and focused ion beam scanning electron microscopes) into the DICOM standard.

As mentioned by the authors, the application and analysis of different image encoding formats is an area of future study that should be pursued, in order to achieve a broad sharing of long-lasting knowledge, and their attempt follows this perspective.

In my opinion, the quality of the paper is high in terms of perspective and data-sharing implementation, average in terms of readibility (punctuation should be revised, for example between line 91-101) and innovation.

The description of the algorithm procedure is accessible even for broad audience. Figure 4 and 5 however, where the conversion pipeline diagram is presented, in my opinion should have a bigger size.

Reviewer 2 Report

In this manuscript, the authors try improving the interoperability of confocal, whole-slide, and FIB-SEM microscope images through DICOM protocol. Standardization of scientific data is very important for application of big data via machine learning and so on. DICOM can be applied to the standardization, because it has been already used in medical area. 

Although I am afraid that only data specialists in this area pay attention to the manuscript, the methodology is publishable in this issue. 

But, there are minor comments as follows. 

To clarify the using DICOM protocol, "decomization" can be changed to "DICOMization" or defined also in the abstract. 

In Fig. 8, the red and glay images from the .lif file are displayed in the channel 4 and 3, respectively. These should be rearranged conversely, because the other red images are displayed in the channel 3. 

In Refs. 3, 7, 18, 21, and 22 from J. Pathol. Inform., the article numbers are missing. These may be 27, 37, 12, 20, and 15, respectively. Please check them. 

Reviewer 3 Report

Gupta et al. report in this article, with titled "Dicomization of Proprietary Files obtained from Confocal, Whole-Slide, and FIB-SEM Microscope Scanners" and submitted to the special issue "Digital Healthcare Leveraging Edge Computing and the Internet of Things", the study on an automated dicomization pipeline in order to transform distinct proprietary microscope images from CLSM, FIB-SEM, and WSI scanners into DICOM standards having biological information within their metadata. The system feasibility and performance were evaluated with 15 distinct proprietary modalities, stacked WSI samples included. The normalized objects were stored through standard communications in the Dicoogle open-source archive. The authors claim that the reported results demonstrated an accurate methodology, which can be used in production. Overall, the work is well-investigated and is very interesting for the biomedical imaging research field. Hence, it could be suitable to be published in the journal Sensors.

Some points and revisions to be addressed to improve the article:

1) In my opinion the 2.2. point is too long and should be simplified. For example, the sentence starting in line 103 "The microscope is an optical..." is too obvious and should be removed.

2) For a proper comparison, Figure 3 shoud show the same image (a, b and c) for the three types of MI modalities.  

3) There are some words of Figures 4 and 5 which are very hard to read because they are too small. Please, rewrite them in a larger size. Besides, in order to make it easier to the readers, Figure 5 should be split in two figures.

4) Figure 9 should be sent to the supplementary material and with a bigger format.

5) The number of quoted references should be increased and the footnotes, concerning six web links, must be transferred to the references section.
